# Intragastric Safflower Yellow Alleviates HFD Induced Metabolic Dysfunction-Associated Fatty Liver Disease in Mice through Regulating Gut Microbiota and Liver Endoplasmic Reticulum Stress

**DOI:** 10.3390/nu15132954

**Published:** 2023-06-29

**Authors:** Wenjing Hu, Xiaorui Lyu, Hanyuan Xu, Xiaonan Guo, Huijuan Zhu, Hui Pan, Linjie Wang, Hongbo Yang, Fengying Gong

**Affiliations:** Key Laboratory of Endocrinology of National Health Commission, Department of Endocrinology, Peking Union Medical College Hospital, Chinese Academy of Medical Science and Peking Union Medical College, Beijing 100730, China; hwjhuwenjing@163.com (W.H.); lvxiaorui0219@foxmail.com (X.L.); jadef21@foxmail.com (H.X.); guoxiaonan199708@163.com (X.G.); shengxin2004@163.com (H.Z.); panhui20111111@163.com (H.P.); eileenwood@163.com (L.W.); hongbo.yang7@gmail.com (H.Y.)

**Keywords:** safflower yellow (SY), metabolic-associated fatty liver disease (MAFLD), gut microbiota, lipogenesis, endoplasmic reticulum stress (ERS)

## Abstract

The gut microbiota was reported to play a significant role in the progression of the metabolic associated fatty liver disease (MAFLD). Our recent study suggested that gastrointestinal tract and liver were important targets mediating the anti-obesity effects of intragastric safflower yellow (SY). Therefore, our present study aims to investigate the effect of intragastric SY on MAFLD and possible mechanism. DIO mice were treated with 125 mg/kg/d SY for 12 weeks by gavage. We found intragastric SY significantly slowed weight gain of body, reduced the food intake and liver weight, improved hepatic steatosis, liver function and glucose metabolism in DIO mice. The comparison between OGTT and IPGTT illustrated OGTT produced a better improvement of glucose tolerance after SY treatment. We also found intragastric SY significantly increased the energy expenditure and locomotor activity of DIO mice. SY obviously decreased the expression of lipogenesis-associated and ERS-related genes in liver of DIO mice and PA-induced MAFLD hepatocyte model. Gut microbiota analysis demonstrated intragastric SY apparently changed the diversity and composition of gut microbiota of DIO mice. Further function prediction analysis indicated that gut microbiotas in SY-treated mice was positively related with energy metabolism, lipid metabolism and endocrine system. Intragastric SY has a significant therapeutic effect on MAFLD, which is mediated partly by modulating gut microbiota and improving liver ERS.

## 1. Introduction

Metabolic associated fatty liver disease (MAFLD), used to be named as nonalcoholic fatty liver disease (NAFLD), is characterized by excessive fat deposition in hepatocytes, with overweight/obesity, type 2 diabetes mellitus (T2DM) or another metabolic dysfunction [1,2]. MAFLD is a clinical pathological syndrome that comprises a continuum of histological abnormalities, including steatohepatitis, hepatofibrosis and cirrhosis [3]. The population prevalence of MAFLD was about 25% worldwide, which was considered as the most common chronic liver disease [4]. MALFD not only affects the quality of human life and life expectancy, but also brings a heavy economic burden [5,6,7]. However, pharmacological therapy specifically for MAFLD approved by FDA is still not available at present, lifestyle modifications including diet and exercise are the primary recommendations for patients with MAFLD [1,8].

*Carthamus tinctorius* L. is a traditional Chinese medicine, which has been mainly used for the treatment of cardiovascular and cerebrovascular diseases in Asian countries for over 2000 years [9]. Safflower yellow (SY) is the major water-soluble component of the flower of *Carthamus tinctorius* L., and hydroxysafflor yellow A (HSYA) is its main bioactive constituent [10]. Our previous studies firstly found that intraperitoneal administration of SY/HSYA significantly slowed the body weight gain, reduced the fat mass, ameliorated other indicators of glucose and lipid metabolism in diet-induced obese (DIO) mice [11,12]. Meanwhile, improved hepatic steatosis and liver function were also observed in SY/HSYA-treated mice from Adamska I et.al and our previous study [13,14], indicating that SY/HSYA might be a potential drug for the treatment of MAFLD.

There are tens of thousands of gut microbiota inhabiting in the gastrointestinal tract and playing an important role in metabolic health of host [15]. Gut microbiota were reported to take part in multiple physiological functions of host, including the maintenance of the intestinal mucosal barrier, bile acid metabolism, the absorption of lipid and vitamins and protection against pathogen invasion [16]. It was demonstrated that some bacterial patterns were the contributing factors in the onset of MAFLD by gut fecal microbial transplantation into germ-free animal models [17,18]. The imbalance of gut microbiota might increase the intestinal permeability and contribute to the liver exposed to exogenous harmful substance, leading to the occurrence and development of hepatic steatosis and fibrosis [19]. Besides, it was observed that the relative abundance of Firmicutes and Actinomycetes was increased in MAFLD animal models, while the relative abundance of Bacteroidetes, Verrucomicrobia, Thermus, Fusobacteria and Leptoshpaeria was reduced [20,21]. Interestingly, some probiotics supplementation was showed to ameliorate hepatic steatosis in MALFD animal models [22,23,24].

Dysfunction of hepatocyte endoplasmic reticulum (ER) homeostasis and lipid accumulation have been recognized to be involved in the MAFLD pathophysiology [25,26]. ER is an essential subcellular compartment responsible for the synthesis and folding of proteins and the change of ER homeostasis would lead to the unfolded protein response, which may exacerbate hepatocytic ballooning [27]. There were studies showing the improvement of ERS provided comparable efficacy in improving diet-induced MAFLD [28,29]. Reducing lipid production and accumulation was also identified as an effective way to reduce hepatic lipid load and ameliorate MAFLD [30].

Our previous studies found that intraperitoneal injection of SY/HSYA could improve hepatic steatosis and liver function of DIO mice [11,12]. However, injection treatment for anti-obesity is obviously unfavorable in clinical application. It was interesting that Liu’s study showed that intragastric HSYA, the main bioactive monomer of SY, could alleviate hepatic steatosis and improve liver function of DIO mice [31]. It was still unclear whether SY, a Chinese FDA approved drugs for the treatment of cardiovascular and cerebrovascular diseases, had the similar effects. Therefore, in our present study, we tried to investigate the effect of intragastric SY on high fat diet (HFD-induced MAFLD and the possible mechanism based on gut microbiota and liver tissue and PA-induced MAFLD hepatocyte model.

## 2. Materials and Methods

### 2.1. Preparation of SY

SY powder was isolated from the water-soluble component of the flower of *Carthamus tinctorius* L., which was provided by Zhejiang Yongning Pharmaceutical Co., Ltd. (Taizhou, Zhejiang, China). The result of the high-performance liquid chromatography (HPLC) analysis of SY was presented in Appendix A. The contents of HSYA in the SY is 87.5%, and the purity of HSYA was 94.2%. SY powder was dissolved in sterilized deionized water to prepare the stock solutions. Working solutions for cell experiments were diluted by complete cell culture medium.

### 2.2. Animal Experiments

Male 7-week-old C57BL/6J mice were purchased from Beijing Vital River Laboratory Animal Technology Co., Ltd. (Beijing, China), and housed in the standard pathogen-free (SPF) environment with free access to water and food. After 1 week of acclimation, mice were randomly assigned to a standard food (SF) group (*n* = 10) and a high-fat diet (HFD) group (*n* = 20). For HFD group, mice were fed with food (45% kcal fat, H10045, Beijing HFK Bioscience Co., Ltd., Beijing, China) for 8 weeks to establish DIO model. Then, HFD-fed mice were randomly divided into HFD group and HFD-SY group (*n* = 10 in each group). Mice in HFD-SY group were treated with 125 mg/kg/d SY by gavage for 12 weeks, while mice in SF and HFD group were treated with equal volume of saline in the same way. Body weight of mice was recorded twice a week, and food intake was recorded three times a week.

### 2.3. Oral Glucose Tolerance Test (OGTT), Intraperitoneal Glucose Tolerance Test (IPGTT) and Intraperitoneal Insulin Tolerance Test (IPITT)

Oral glucose tolerance test (OGTT), intraperitoneal glucose tolerance test (IPGTT) and intraperitoneal insulin tolerance test (IPITT) were conducted after 8 weeks of SY intervention. For OGTT and IPGTT, mice had fasted overnight (about 15 h), and then were treated with 50% glucose (2 mg/g) by gavage or intraperitoneal injection. For IPITT, after 5 h of fasting, mice were treated with insulin (0.7 IU/kg, Novolin R, Novo Nordisk, Denmark) by intraperitoneal injection. Blood glucose levels of mice were measured by using a glucometer (Roche, Basel, Switzerland) from the tail at 0, 30, 60, 90 and 120 min. The area under the curve (AUC) was calculated by using trapezoidal integration method.

### 2.4. Assessment of Indirect Calorimetry

After 10 weeks of intervention, five mice in each group were randomly assigned to the assessment of indirect calorimetry by Promethion Metabolic Cage system (Sable Systems International, Las Vegas, NV, USA) according to the instruction manual. The housing conditions were the same with those of SPF environment as described in Section 2, and SY intervention was carried out normally during the experiment. After 1 week of acclimation, mice were put into metabolic cages for 3 days to monitor oxygen consumption (VO_2_), carbon dioxide production (VCO_2_), respiratory exchange rate (RER), energy expenditure (EE), pedometers, wheelmeters and allmeters. The relatively stable data of 24 h during the experiment were used for analysis.

### 2.5. Samples Collection and Biochemical Measurements

After 12 weeks of SY intervention, the mice were fasted overnight and intraperitoneally injected with 500 mg/kg 2,2,2-tribromoethanol (Sigma-Aldrich, St. Louis, MO, USA) for anesthesia. The epididymal white adipose tissue (WAT), perirenal WAT, subcutaneous WAT and liver were collected and weighed and then frozen at −80 °C for following experiments. The blood samples were centrifuged at 3000 rpm for 10 min at 4 °C to obtain serum samples. Serum biochemical parameters, including hepatic and renal function markers, blood lipids markers and blood glucose, were measured using routine automated laboratory methods as described in our previous study [32]. The tissue samples were stored in Clinical Biobank, Peking Union Medical College Hospital, Chinese Academy of Medical Sciences (Beijing, China) until they were used.

### 2.6. Histopathological Analysis of Liver and Epididymal WAT

A part of the epididymal WAT and liver tissue were fixed in 10% paraformaldehyde. Dehydration, embedding, slicing and Hematoxylin-Eosin (H&E) staining were carried out according to the standard protocol. Then histological images of epididymal WAT and liver tissues were obtained by using a microscope with digital camera (Nikon DS-U3, Tokyo, Japan) at 100× or 200× magnification (scale bars, 50 or 100 µm). The MAFLD score were quantified by a MAFLD scoring system as described previously [33]. Briefly, it included steatosis and inflammation. The steatosis scores were evaluated according to hypertrophy (cellular enlargement more than 1.5 times the normal hepatocyte diameter), macro vesicular steatosis and micro vesicular steatosis, every item was graded, based on the percentage of the total area affected, into the following categories: 0 (<5%), 1 (5–33%), 2 (34–66%) and 3 (>66%). The scores of inflammations were evaluated by counting the number of inflammatory foci (a focus was defined a cluster, not a row, of ≥5 inflammatory cells) per field at 100× magnification. Five different fields at 100× magnification of each sample were counted and averaged according to the following categories: 0 (<0.5 foci), 1 (0.5–1.0 foci), 2 (1.0–2.0 foci), 3 (>2.0 foci). In addition, liver index was calculated by the following equation: Liver index = Liver weight (g)/ Body weight (g).

### 2.7. Gut Microbiota Analysis

Microbial DNA was extracted from the cecum contents of mice using Cetyltrimethyl Ammonium Bromide (CTAB) method. Briefly, CTAB extraction buffer could form a complex with proteins and polysaccharides in high ionic strength solution, but it cannot precipitate nucleic acids. Therefore, DNA could be extracted by this way as previously described [34]. The V1-V9 regions of the bacteria 16S rRNA gene were amplified using the Forward primer 5′-AGAGTTTGATCCTGGCTCAG-3′ and the Reverse primer 5′-GNTACCTTGT TACGACTT-3′. Amplicons were purified by using AMpure PB magnetic beads, quantified by using Qubit 2.0 (Invitrogen, Carlsbad, CA, USA), and then the sequencing library was generated on the PacBio platform (Pacific Biosciences, Menlo Park, CA, USA). Clean Reads were clustered into operational taxonomic units (OTUs) with the 97% similarity using Uparse software (Version 7.0.1001). Mothur method and SSUrRNA database of SILVA were used to annotate taxonomic information. Venn diagram, alpha and beta diversity analyses, Principal co-ordinates analysis (PCoA) and Linear discriminate analysis effect size (LEfSe) were performed by QIIME software (Version 1.9.1) and R software (Version 2.15.3). PICRUSt (Phylogenetic Investigation of Communities by Reconstruction of Unobserved States) was used to predict the function of the gut microbiota. Correlation analyses of gut microbiota and metabolic profiles were performed by R software (Version 2.15.3).

### 2.8. Cell Culture and Cell Experiments of HepG2 Cells

HepG2 cells were purchased from the Cell Resource Center, Institute of Basic Medical Sciences, Chinese Academy of Medical Sciences (Beijing, China) and were cultured in minimum essential medium with Earle’s Balanced Salts (MEM, HyClone, Logan, UT, USA) supplemented with 10% fetal bovine serum (FBS, Gbico, Grand Island, NE, USA), 2% HEPES, 0.1 mM non-essential amino acids (NEAA) and 1% antibiotics (100 μg/mL streptomycin and 100 U/mL penicillin) at 37 °C in a 5% CO_2_ incubator (Thermo, Waltham, MA, USA). HepG2 cells were pretreated with 10, 50 and 100 mg/L SY for 12 h, and then treated with 400 μM palmitic acid (PA) for 12 h to induce lipid accumulation, following further experiments.

### 2.9. Cell Viability Assay

The effect of PA on cell viability of HepG2 cells was determined by Cell Counting Kit-8 (CCK8, MedChem Express, HY-K0301, Monmouth Junction, NJ, USA) according to the instruction manual. HepG2 cells were plated at a density of 5 × 103 cells per well in 96-well plates. After 24 h, HepG2 cells were treated with 0, 200, 400, 600, 800 and 1000 μM PA for 12 h. The culture medium was replaced with fresh culture medium and mixed with 10 μL CCK8 reagent. After 1h incubation, the absorbance at 450 nm was measured using a microplate reader (Thermo, Waltham, MA, USA).

### 2.10. Oil Red O Staining

The lipid accumulation in HepG2 cells were visualized by Oil Red O staining using the Oil red O staining kit (ORO, Sigma, St. Louis, MO, USA). In brief, HepG2 cells were rinsed with PBS for 3 times, fixed with 60% isopropanol for 5min, and stained with Oil Red O dye for 15 min. After rinsed with sterilized distilled water for 5 times, the photographs of HepG2 cells were captured by a digital camera (Nikon) at 100× magnification. Then oil red O dye was extracted by 60% isopropanol and the absorbance at 490 nm was measured using a microplate reader as detailed in our previous study [35].

### 2.11. Reverse Transcription-Quantitative Polymerase Chain Reaction (RT-qPCR)

Total RNA was extracted from liver tissue and HepG2 cells using Total RNA kit II (Omega Biotek, Norcross, GA, USA). The PrimeScript™ RT reagent Kit with gDNA Eraser (TaKaRa, Kyoto City, Japan) was used to perform the reverse transcription. The expressions of uncoupling protein 1 (*Ucp1*), peroxisome proliferative activated receptor, gamma, coactivator 1 alpha (*Ppargc1a*), peptidylprolyl isomerase A (*Ppia*), acetyl CoA carboxylase (*Acc*), sterol regulatory element binding transcription factor 1 (*Srebf1*), fatty acid synthetase (*Fasn*), stearyl CoA desaturase 1 (*Scd1*), tumor necrosis factor (*Tnf*), interleukin 1b (*Il 1b*), peroxisome proliferator activated receptorα (*Ppara*), carnitine palmitoyl transferase 1α (*Cpt1a*), activating transcription factor 4 (*Atf4*), activating transcription factor 6 (*Atf6*), DNA-damage inducible transcript 3 (*Ddit3*), heat shock protein 5 (*Hspa5*), X-box binding protein 1 (*Xbp1*) and *Xbp1s* were detected by using TB Green^®^ Premix Ex TaqTM II (TaKaRa, Japan) in the ABI7500 PCR system (Applied Biosystems, San Francisco, CA, USA). *Ppia* and *GAPDH* were used for the normalization of above target genes. The primers of these gene were listed in Appendix A. The 2^−ΔΔCt^ method was used to calculate the relative expressions of target genes.

### 2.12. Statistical Analysis

All data were expressed as mean ± standard error. All the statistical analysis was performed by GraphPad Prism Software (Version 9.0 for MacBook, GraphPad Prism Inc., San Diego, CA, USA). If the data conformed to the normal distribution and the variance was homogeneous, One-way ANOVA was used for data analysis. If not, Kruskal-Wallis was performed. *p* < 0.05 was considered statistically significant.

## 3. Results

### 3.1. Intragastric SY Ameliorates HFD-Induced Obesity and Systemic Metabolic Dysfunction

The graphical protocol of animal experiment was summarized as Figure 1A. In the setting of HFD feeding for 12 weeks, significant differences were observed in murine body weight, food intake, adiposity, glucose tolerance, insulin sensitivity, serum lipid and glucose level between the DIO control and SF control mice (Figure 1B–M, *p* < 0.05). The intragastric SY conspicuously reduced the body weight gain and body weight gain percentage of DIO mice, equivalent to 56.0% and 56.9% of HFD group, respectively (Figure 1C, *p* < 0.05). The food intake of SY-treated mice was continuously lower than DIO control mice, which was statistically significant at 2nd, 3rd, 5th and 6th weeks (Figure 1D, *p* < 0.05). Intragastric SY also led to a lower WAT percentage, which was 72.3% of DIO control mice (Figure 1E, *p* < 0.05), with smaller epididymal adipocyte sizes (Figure 1M). Moreover, a faster glucose disposal in the OGTT and IPGTT as well as a greater responsiveness to insulin in IPITT than DIO control mice was detected after intragastric SY treatment (Figure 1F–L, *p* < 0.05). Consistently, the AUC of OGTT and IPITT of SY-treated mice reduced to 83.0% and 76.9% of DIO control mice (Figure 1G,L, *p* < 0.05), respectively, while no significant difference in AUC of IPGTT was observed between above two groups (Figure 1I). We further compared the curve and AUC of OGTT and IPGTT in SF group, HFD group and HFD-SY group. As presented in Figure 1J, the AUC of OGTT in the SF was decreased to 74.6% of the AUC of IPGTT (Figure 1J, *p* < 0.05), suggesting the glucose disposal in the OGTT in SF group was faster than in the IPGTT as we all known. However, the reduction phenomenon was disappeared in HFD group. Interestingly, the reduction phenomenon was reappeared after SY treatment DIO mice, as evidenced that the AUC of OGTT in HFD-SY groups was decreased to 84.7% of the AUC of IPGTT (Figure 1J, *p* < 0.05), indicating that SY treatment has a significant role in gastrointestinal tract. However, intragastric SY had no effect of serum FBG, TG, TC, LDL-c and HDL-c level in DIO mice as shown in Table 1.

### 3.2. Intragastric SY Increases Energy Expenditure of DIO Mice

In our previous study, intraperitoneal treatment of SY was showed to promote the browning of subcutaneous WAT in the DIO mice [11], which was an important way to increase energy expenditure. Therefore, our present study investigated whether intragastric SY affect energy balance of DIO mice. Metabolic cage experiment showed that intragastric SY significantly increased the energy expenditure of DIO mice to 1.1-fold of DIO controls (Figure 2G,H, *p* < 0.05), and an increased trend of VO_2_ (*p* = 0.07) was also observed in SY-treated mice when compared with DIO controls (Figure 2A). Consistently, both the locomotor activity (wheelmeters) and allmeters of SY-treated mice was increased to 1.6-fold of DIO control mice (Figure 2K,L, *p* < 0.05). In addition, a decreased trend of food intake (Figure 2I, *p* = 0.06) was observed in SY-treated mice when compared with DIO control mice. Considering the increase in energy expenditure, we further explored the expression thermogenesis-related gene including *Ucp1* and *Ppargc1a* in subcutaneous adipose tissue (sAT) and brown adipose tissue (BAT). In sAT, no significant increase was observed after SY intervention (Figure 2M,N), while in BAT, the expression of *Ucp1* and *Ppargc1a* increased to 1.7-fold and 1.9-fold of DIO control mice respectively (Figure 2O,P, *p* < 0.05).

### 3.3. Formatting of Mathematical Components

Compared with SF control mice, a 12-week HFD-feeding led to larger liver size and liver weight, increased hepatic lipid accumulation and MAFLD score, accompanied with upregulation of lipogenic genes (Figure 3A–F, *p* < 0.05). Intragastric SY significantly decreased the liver weight and liver index to 81.3% and 82.7% of DIO control mice (Figure 3B, *p* < 0.05), respectively, with smaller liver size (Figure 3A). The serum ALT level in SY-treated mice was also decreased to 43.5% of HFD group (Figure 3C, *p* < 0.05), indicating the improved liver function. In addition, subsequent H&E staining showed a decrease in lipid droplets of hepatocytes in SY-treated mice when compared with DIO control mice (Figure 3D). Consistently, the MAFLD score of mice decreased to 14.8% of DIO control mice after intragastric SY treatment (Figure 3E, *p* < 0.05). The lipogenic gene *Fasn* mRNA levels significantly decreased to 46.8% of DIO control mice after intragastric SY treatment (Figure 3F, *p* < 0.05). In addition, intragastric SY reduced significantly inflammation-related gene *Tnf* and *Il1b* by 60.5% and 47.5% respectively (Figure 3G, *p* < 0.05). Finally, ESR-related gene *Atf4*, *Hspa5* and *Xbp1s* decreased by 38.1%, 49.3% and 46.7% after SY intervention (Figure 3H, *p* < 0.05). However, the expressions of genes associated with fatty acid oxidation have not changed statistically (Figure 3I).

### 3.4. Intragastric SY Alters the Composition of Gut Microbiota in DIO Mice

Since a faster glucose disposal in the OGTT than in the IPGTT was observed in HFD-SY group, which suggest that SY treatment may significantly play a significant role in gastrointestinal tract, the effect of intragastric SY on gut microbiota was further investigated by the full length (V1–V9) 16s rRNA sequencing. As shown in Appendix A, HFD feeding led to the decrease in rank abundance of the OTUs, while intragastric SY rescued the loss of OUTs caused by HFD feeding. Then, differential bacteria in top 10 abundance among the three groups were dissected from phylum levels to the species levels. At the phylum levels, compared with SF group, the relative abundance of Firmicutes was increased in HFD group, while SY significantly reduced its relative abundance to 78.1% of HFD group (Figure 4A, *p* < 0.05). Meanwhile, the relative abundance of Verrucomicrobiota in HFD-SY group increased to 22.6-fold of HFD group (Figure 4A, *p* < 0.05). At the genus levels, the relative abundance of Akkermansia and Bacteroides in SY-treated mice rose to 22.6-and 153-folds of DIO mice, respectively (Figure 4B, *p* < 0.05). At the species levels, the relative abundance of Bacteroides_acidifaciens and Bacteroides_vulgatus were also significantly increased after SY treatment, which were equivalent to 132.6- and 145.7-folds of HFD group, respectively (Figure 4C, *p* < 0.05).

Further assessment of α diversity of gut microbiota indicated that HFD feeding notably impaired the richness of gut microbiota (*p* < 0.05), while SY treatment showed an increasing trend in the richness although no statistically significant differences were observed (Figure 5A). Moreover, the β diversity index demonstrated SY treatment evidently increased the richness of gut microbiota (Figure 5B, *p* < 0.05). The results of PcoA revealed that the gut microbial composition of HFD group and HFD-SY group was obviously separated (Figure 5C). LEfSe analysis was used to explore the unique gut microbiota among three different groups. Notably, phylum Bacteroidetes, class Bcteroidia, order Bacteroidales and family Muribaculaceae were the most overrepresented bacteria in SF group, phylum Firmicutes were mainly enriched in HFD group, while family unidentified_Clostridiales, family Bacteroidaceae, Bacteroides, Butyriciococcus and Bacteroides_intestinalls were the most strongly associated with HFD-SY group (Figure 5D,E). The relationship between the metabolic phenotypes and gut microbiota was further analyzed by the Spearman’s correlation analysis. As shown in Figure 5F, Bacteroides, Bacteroides_acidifaciens and Bacteroides_intestinalls were negatively correlated with food intake, while Firmicutes was positively correlated with WAT percentage of DIO mice. In addition, PICRUSt was used to predict the functional differences of the gut microbiota among three groups. Interestingly, both SF group and HFD-SY group were positively related with energy metabolism, lipid metabolism, and endocrine system, while HFD group were negatively related with the above pathways (Figure 5G). The above results indicated that intragastric SY might increase the energy metabolism and lipid metabolism by influence the composition and function of gut microbiota.

### 3.5. SY Reduces the Expression of Lipogenesis-Associated and ERS-Related Genes in HepG2 Cells

Next, we investigated the effect of different doses of SY on hepatocytes. As shown in Appendix A, the cell viability of HepG2 cells was not significantly decreased until the concentration of PA increased to 0.6 mM/L. Therefore, 0.4 mM/L PA was used to establish MAFLD hepatocyte model in vitro. More and larger lipid droplet were observed in PA-treated HepG2 cells when compared with control cells, accompanied with upregulation of lipogenic genes, indicating the PA-induced MAFLD cell model was well-established (Figure 6A–C, *p* < 0.05). As presented in Figure 6A,B, 10–100 mg/L SY treatment conspicuously decreased the intracellular triglyceride contents of HepG2 cell and the intracellular triglyceride contents decreased to 73.8–80.5% of control cells, respectively (*p* < 0.05). Moreover, similar to the results obtained in liver tissue of SY-treated DIO mice, 10–100 mg/L SY treatment evidently decreased the *SREBP1*, *FASN* and *ACC* mRNA levels to 71.4–78.3% respectively (Figure 6C, *p* < 0.05). As for *SCD1*, there were about 50% reduction after 10–100 mg/L SY treatment. Moreover, the expression of *ATF4*, *DDIT3* and *XBP1*, which were involved in ESR-related genes, also statistically decreased with different concentration of SY treatment (Figure 6E, *p* < 0.05). As for inflammation related genes, there was statistically great reduction after 10–100 mg/L SY intervention in *TNF* (Figure 6D, *p* < 0.05). Although the expression of IL1b showed no significant difference, there was downward trend in 10–100 mg/L SY group (Figure 6D). However, there was no statistical differences in the expression of fatty acid oxidation related genes including *PPARa* and *CPT1a* in HepG2 cells with or without SY treatment (Figure 6F).

## 4. Discussion

Currently, there are still no reliable drug specifically for MAFLD. Our present study systematically investigated the effects of intragastric SY. On the one hand, intragastric SY could reduce bodyweight and improve glucolipid metabolism. We also found its effect on MAFLD, which meant it had significant effects on alleviating hepatic steatosis and improving liver function of DIO mice. Further mechanistic study showed intragastric SY significantly increased energy expenditure, altered the composition of gut microbiota, decreased the expression of ERS related genes and lipogenesis in liver of mice. These findings suggested that intragastric SY may be a potential treatment for MAFLD in the future.

In our previous studies, intraperitoneal injection of SY/HSYA could ameliorate hepatic steatosis and improve liver function in DIO mice [12]. However, injection method is obviously not suitable for clinical application of SY. Interestingly, our present study firstly found that oral SY treatment, a more clinically suitable method, had same effects on improving MAFLD. Consistent with our study, Liu’s study also found that intragastric HSYA, the main bioactive monomer of SY, also remarkedly reduced lipid contents and inflammation in the liver of DIO mice [31]. Both excess hepatic lipid accumulation and ERS are considered to be crucial factors contributing to the pathogenesis of MAFLD [3,36]. Further mechanism investigation in our present study firstly showed that intragastric SY administration significantly decreased the expression of lipogenesis-related gene *Fasn* and ERS-related gene *Atf4*, *Hspa5* and *Xbp1s* which was the potent transcription factor translated from the spliced Xbp1 mRNA in liver tissue of mice. Similarly, in PA-induced MAFLD hepatocyte model, we also found SY could reduce intracellular triglyceride, and lipogenesis-related transcription factors *SREBP1* and its target genes *FASN*, *ACC* and *SCD1*. In addition, the expression of genes involved in ERS like *ATF4*, *DDIT3* and *XBP1* evidently reduced after SY treatment. Consistent with our results, Cheng et al. found that glycoursodeoxycholic acid protected mice from HFD-induced hepatic steatosis and alleviated ERS in livers of high fat diet (HFD)-fed mice [29]. ERS was reported to cause reactive oxygen species (ROS) accumulation and diffusion into the cytoplasm leading to cellular oxidative stress (OS), and OS can induce protein misfolding and ERS [37]. Moreover, our previous study showed that intraperitoneal injection of SY/HSYA could increase the expression of antioxidant enzymes and superoxide dismutase (SOD) activities in the liver tissue of DIO mice and HepG2 cells [12]. All these findings suggest that SY has significantly direct effects on alleviating hepatic steatosis and improving liver function in DIO mice.

Our previous study indicated intragastric SY/HSYA, rather than an intraperitoneal injection, notably decreased serum *Gip* levels and *Gip* staining in the small intestine in diet-induced obese (DIO) mice [38], suggesting intragastric intervention played an important role in gut. Given the important role of gut microbiota plays in the occurrence and development of MAFLD and the potential link between the effects of SY and gastrointestinal tract, the potential contributions of gut microbiota to the benefits of SY treatment on MAFLD were investigated in our present study. The results showed that intragastric SY treatment increased the relative abundance of Verrucomicrobia, while decreased the relative abundance of Firmicutes at the phylum levels. Verrucomicrobia was considered to be a potential prebiotic. Firmicutes was one of two dominant bacterial divisions, which increased in both obesity and MAFLD mice [20,39]. At the genus levels, intragastric SY treatment increased the relative abundance of Bacteroides and Akkermansia. Bacteroides, the main genus of the Bacteroidetes phylum, was conspicuously reduced in gastrointestinal tract of obese and MAFLD population [20,40]. Akkermansia, a mucin-degrading bacterium, is considered to be a probiotic, which plays a significant role in prevention and treatment of obesity and related metabolic diseases [41,42]. It was reported that Akkermansia intervention could ameliorate hepatic steatosis, reduce inflammatory reaction and improve glucose metabolism in DIO mice [43,44]. Consistent with our findings, previous study found that HSYA treatment might promote the expression of the tight junction proteins ZO-1 in colon through increasing the relative abundance of Akkermansia, contributing to the intestinal integrity [31,45]. Besides, it was also reported that liraglutide might reduce the liver TG contents and improve the liver function in db/db mice through increasing the relative abundance of Akkermansia [46]. In addition, supplementation with Akkermansia muciniphila in overweight/obese patients could improve insulin sensitivity, reduce insulinemia, plasma TC, fat mass as well as hip circumference and decrease the levels of the relevant blood markers for liver dysfunction and inflammation [47]. At species levels, intragastric SY treatment increased the relative abundance of Bacteroides_acidifaciens and Bacteroides_vulgatus. MALFD rats treated with antibiotic showed severe steatohepatitis and gut microbiota alteration, while the MAFLD phenotypes could be partly reversed by increasing the relative abundance of probiotic such as Bacteroides_acidifaciens [48]. Moreover, colonisation by Bacteroides_acidifaciens in germ-free mice was demonstrated to promote the acetate production, which played important roles in protecting against MAFLD development [49]. Bacteroides_acidifaciens-reconstituted mice were also more resistant to ConA-induced liver injury and alcoholic liver injury [50]. Consistently, it was reported that Bacteroides_acidifaciens might, at least partly, mediate the beneficial effects of garlic organosulfur compounds, broccoli microgreens juice, U. pinnatifida and Yijin-Tang on reducing body weight and liver weight, alleviating insulin resistance, and ameliorating hepatic steatosis through promoting the production of short-chain fatty acids and reducing lipopolysaccharide (LPS) levels in DIO and MAFLD mice [51,52,53,54]. As to Bacteroides_vulgatus, only one study reported that it was decreased in T2DM patients [55]. Further gut fecal microbial transplantation in germ free animal models was needed to explore the effect of Bacteroides_vulgatus on MAFLD.

Physical activity is a critical factor that influence the energy expenditure of body [56]. It was reported that increasing energy expenditure through aerobic exercise could reduce the liver fat of obese or MAFLD patients [57]. Our previous study also showed that nuciferine, the main aporphine alkaloid component in lotus leaf, obviously ameliorated hepatic steatosis and decreased MAFLD scores in DIO mice through increasing energy expenditure [58]. One striking finding of our study was that intragastric SY was firstly found to increase energy expenditure and locomotor activity of DIO mice. Further function prediction of gut microbiotas showed that gut microbiotas of mice in HFD-SY group was positively, while gut microbiotas of mice in HFD group was negatively related with energy metabolism. These findings indicate that intragastric SY might increase the energy expenditure in DIO mice by modulating the gut microbiotas, therapy improving the MAFLD.

It was well established that hypocaloric diet was the cornerstone of treatment for MAFLD [57,59]. Our study found that intragastric SY obviously reduced the food intake of DIO mice during the intervention, which has also been found in our previous results [38]. Our further correlation analysis of the metabolic phenotypes and gut microbiota showed that Bacteroides, Bacteroides_acidifaciens and Bacteroides_intestinalls were negatively correlated with food intake of DIO mice, while their abundance was significantly increased in SY-treated mice. These findings indicate that intragastric SY may reduce the food intake of DIO mice by modulating the abundance of gut microbiotas.

Our present study firstly found that gut microbiota played a significant role in anti-MAFLD effect of intragastric SY by the full-length (V1–V9) 16S rRNA sequencing, which could detect longer regions of bacteria 16S rRNA than only V4 16S rRNA sequencing used in Liu’s study [31] and conduct analysis at species levels. Meanwhile, other effects including the increase of EE and the improvement of ERS as well as lipogenesis played an important role in ameliorating MAFLD. Even so, fecal microbiota transplantation experiment in germ-free mice was needed to verify the role of gut microbiota on the anti-MAFLD effect of SY. Additionally, more advanced sequencing methods, such as metagenomics, were need to further investigate the functions of differential gut microbiota between HFD group and HFD-SY group.

## 5. Conclusions

Intragastric SY could attenuate MAFLD in high-fat diet fed mice, which is mediated, at least partly, through increasing energy expenditure, modulating gut microbiota and improving directly the ESR as well as lipogenesis of liver.

## Figures and Tables

**Figure 1 nutrients-15-02954-f001:**
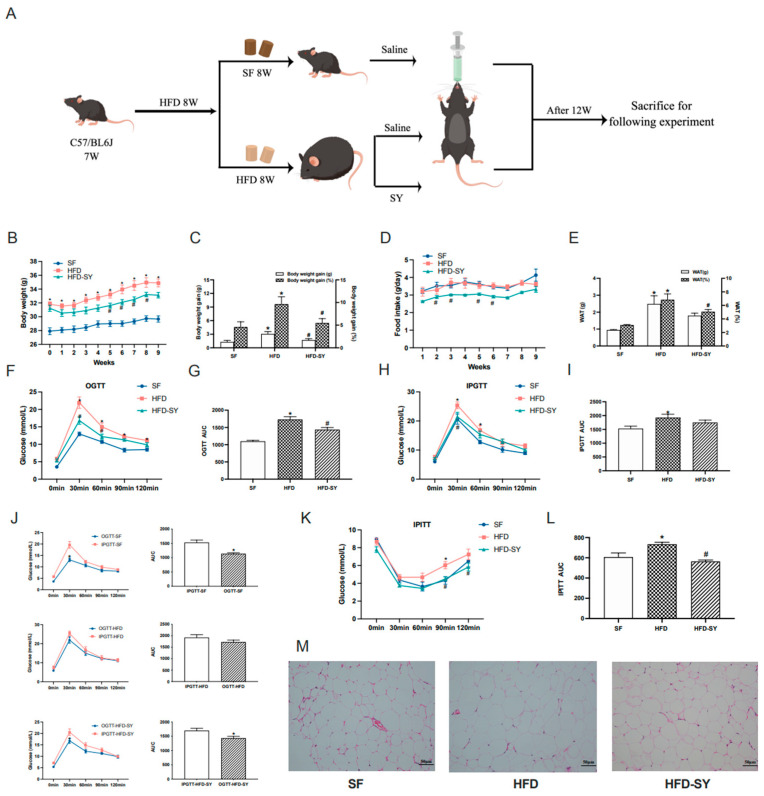
Intragastric safflower yellow (SY) ameliorated diet-induced obesity (DIO) and metabolic dysfunction in mice. DIO mice were treated with SY by gavage for 12 weeks. (**A**) Graphical protocol of animal experiment. (**B**) The body weight of mice from beginning to 9th week. (**C**) The body weight gains and body weight gain percentage of mice in 9th week. (**D**) Food intake of mice from beginning to 9th week. (**E**) The WAT (g) and WAT (%) of mice in 12th week. (**F**) The oral glucose tolerance test (OGTT), (**H**) intraperitoneal oral glucose tolerance test (PIGTT) and (**K**) the intraperitoneal insulin tolerance test (IPITT) in 8th week. (**G**,**I**,**L**) Areas under the curve (AUCs) of the OGTT, IPGTT and IPITT. (**J**) the comparison of the AUCs of OGTT and IPGTT. (**M**) H&E staining images of epididymal WAT sections of mice at ×200 magnification (scale bars, 50 μm). The data were represented as the mean ± S.E.M. (*n* = 10 in each group). *: vs. SF group, *p* < 0.05; #: vs. HFD group, *p* < 0.05.

**Figure 2 nutrients-15-02954-f002:**
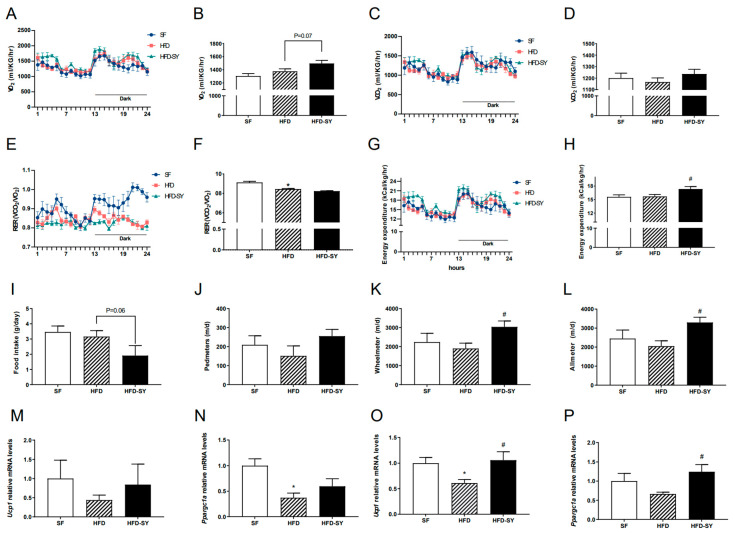
Intragastric SY increased the energy expenditure of DIO mice. Mice were treated with SY by gavage for 10 weeks, then (**A**,**B**) Oxygen consumption (VO_2_) of mice, (**C**,**D**) CO_2_ production (VCO_2_) of mice, (**E**,**F**) Respiratory exchange ratio (RER) of mice, (**G**,**H**) Energy expenditure (EE) of mice, (**I**) Food intake of mice, (**J**) Pedmeters, (**K**) Wheelmeters and (**L**) Allmeters of mice, (**M**,**N**) The mRNA levels of uncoupling protein 1 (*Ucp1*), peroxisome proliferative activated receptor, gamma, coac-tivator 1 alpha (*Ppargc1a*) in subcutaneous adipose tissue, (**O**,**P**) The mRNA levels of *Ucp1* and *Ppargc1a* in brown adipose tissue. The data were represented as the mean ± S.E.M. (*n* = 5 in each group). *: vs. SF group, *p* < 0.05; #: vs. HFD group, *p* < 0.05.

**Figure 3 nutrients-15-02954-f003:**
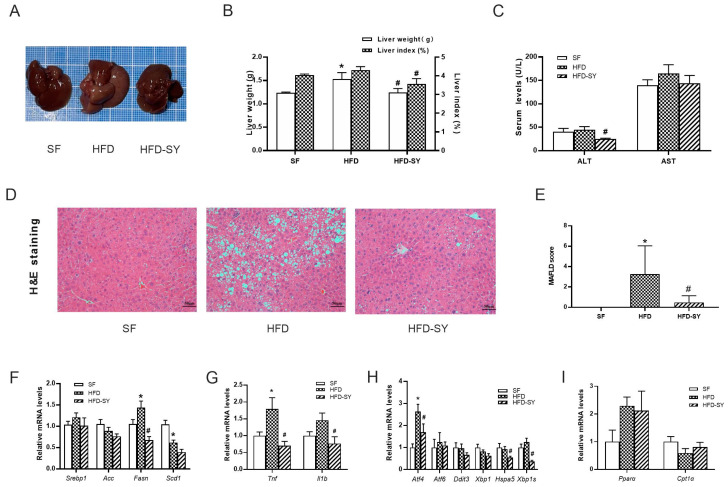
Intragastric SY improved the hepatic steatosis and reduced the expression of lipogenesis-related genes in liver of DIO mice. Mice were treated with SY by gavage for 12 weeks. (**A**) The picture of the whole liver. (**B**) The liver weight (g) and liver index (%) of mice. (**C**)The serum ALT and AST levels of mice. (**D**) H&E staining images of liver sections of mice at ×100 (scale bars, 50 μm). (**E**) The MAFLD score of mice. (**F**–**I**) The mRNA levels of sterol-regulatory element binding proteins (*Srebp1*), acetyl CoA carboxylase (*Acc*), fatty acid synthetase (*Fasn*), stearyl CoA desaturase 1 (*Scd1*), tumor necrosis factor (*Tnf*), interleukin 1b (*Il 1b*), activating transcription factor 4 (*Atf4*), activating transcription factor 6 (*Atf6*), DNA-damage inducible transcript 3 (*Ddit3*), X-box binding protein 1 (*Xbp1*), heat shock protein 5 (*Hspa5*) and *Xbp1s*, peroxisome proliferator activated receptorα (*Ppara*) and carnitine palmitoyl transferase 1α (*Cpt1a*) in liver of mice. The data were represented as the mean ± S.E.M. (*n* = 10 in each group of animal experiments). *: vs. SF group, *p* < 0.05; #: vs. HFD group, *p* < 0.05.

**Figure 4 nutrients-15-02954-f004:**
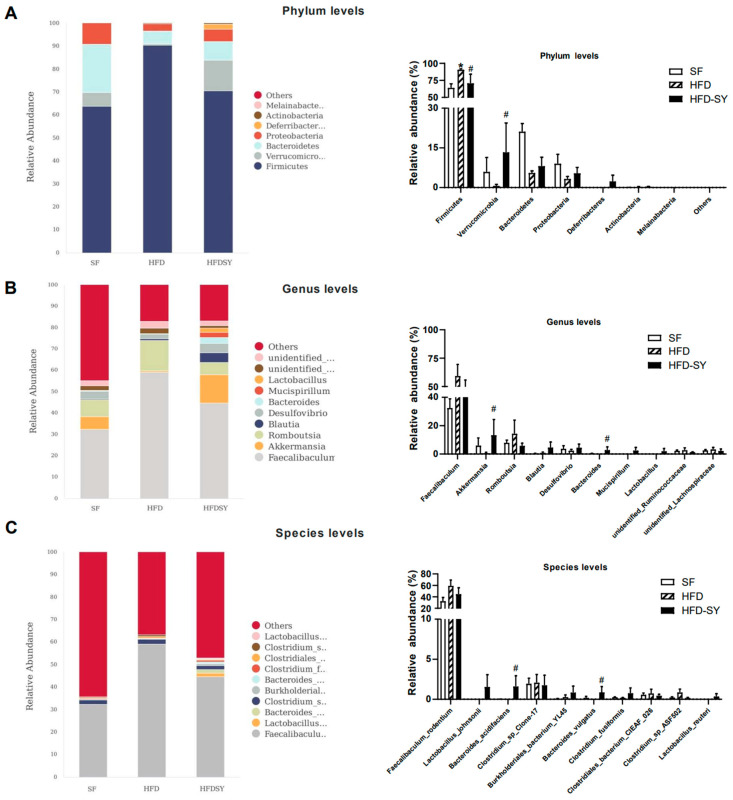
Intragastric SY altered the composition of gut microbiota in DIO mice. Mice were treated with SY by gavage for 12 weeks. (**A**) Differential bacteria in top 10 abundance at the phylum levels. (**B**) Differential bacteria in top 10 abundance at the genus levels. (**C**) Differential bacteria in top 10 abundance at the species levels. The data were represented as the mean ± S.E.M. (*n* = 5 in each group). *: vs. SF group, *p* < 0.05; #: vs. HFD group, *p* < 0.05.

**Figure 5 nutrients-15-02954-f005:**
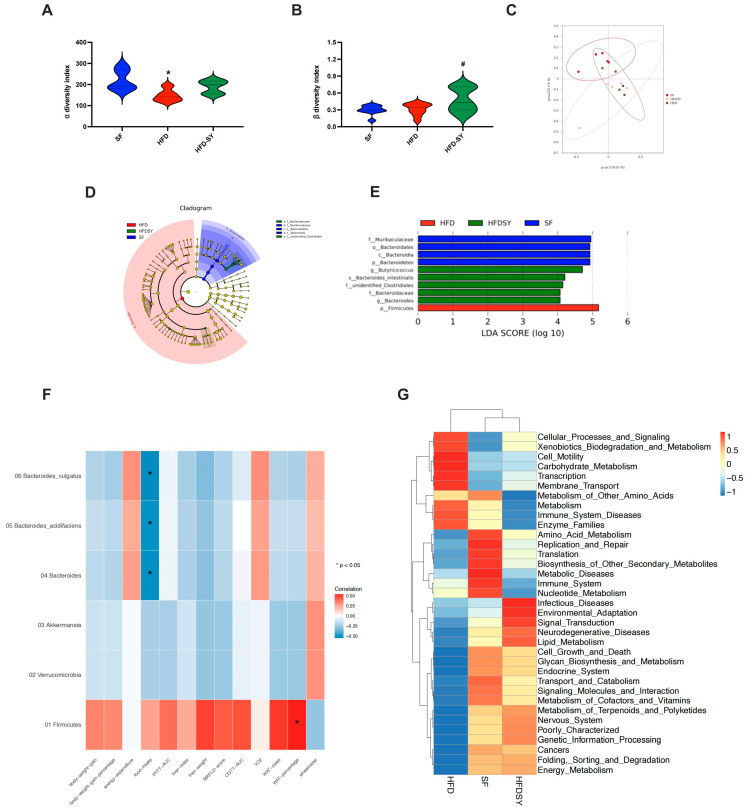
Intragastric SY changed the function of the gut microbiota of DIO mice. Mice were treated with SY by gavage for 12 weeks. (**A**) Violin plot of α diversity index; (**B**) Violin plot of β diversity index; (**C**) Principal co-ordinates analysis (PcoA) plot; (**D**) Linear discriminate analysis effect size (LEfSe) analysis. (**E**) Association map of metabolic phenotypes and differential gut microbiota. (**F**) Heatmap of the predicted function of the gut microbiota (*n* = 5 in each group). (**G**) Heatmap of the correlation of metabolic phenotype and differential gut microbiota (*n* = 5 in each group). The data were represented as the mean ± S.E.M. (*n* = 5 in each group). *: vs. SF group, *p* < 0.05; #: vs. HFD group, *p* < 0.05.

**Figure 6 nutrients-15-02954-f006:**
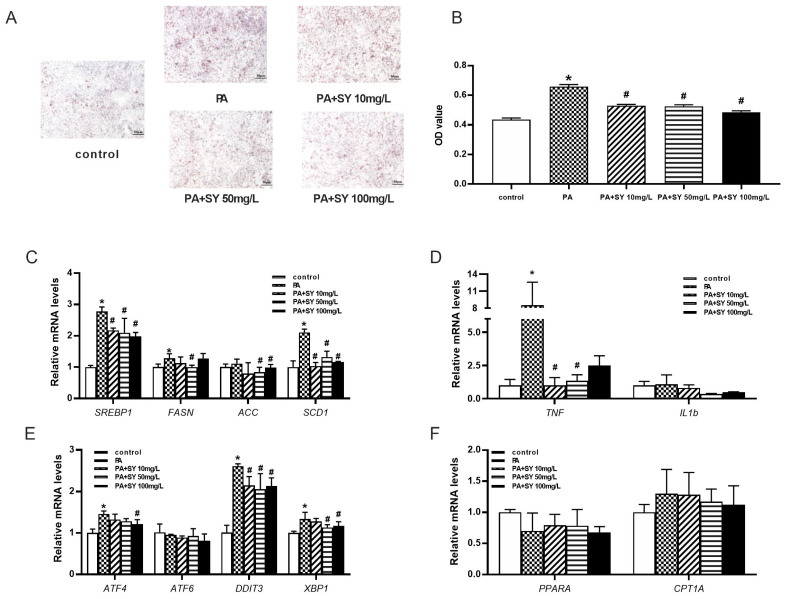
The effect of different dose of SY on the expression of fatty acid-related and endoplasmic reticulum stress-related genes in the PA-induced MAFLD cell model. (**A**) The oil red O (ORO) staining of HepG2 cells. (**B**) the quantify of ORO dye. (**C**–**F**) The mRNA level of *SREBP1*, *FASN*, *ACC*, *SCD1*, *TNF*, *IL1b*, *ATF4*, *ATF6*, *DDIT3*, *XBP1*, *PPARA* and *CPT1A* in HepG2 cells. The data were represented as the mean ± S.E.M. (*n* = 10 in each group of animal experiments, *n* = 3 in each group of cell experiments). *: vs. SF group or control cells, *p* < 0.05; #: vs. HFD group or PA-treated group, *p* < 0.05.

**Table 1 nutrients-15-02954-t001:** Effects of intragastric SY on serum biochemical parameters of DIO mice.

	SF	HFD	HFD-SY
Cr (μmol/L)	12.08 ± 0.36	14.40 ± 0.35	13.60 ± 0.59
Glu (mmol/L)	3.47 ± 0.19	5.55 ± 0.28 *	5.36 ± 0.36
TC (mmol/L)	3.39 ± 0.03	4.96 ± 0.12	4.43 ± 0.57
TG (mmol/L)	0.17 ± 0.03	0.16 ± 0.01	0.12 ± 0.02
HDL-c (mmol/L)	1.44 ± 0.03	1.75 ± 0.11	1.84 ± 0.06
LDL-c (mmol/L)	0.22 ± 0.01	0.42 ± 0.03 *	0.36 ± 0.05

TC, total cholesterol; TG, triglycerides; LDL-c, low density lipoprotein-cholesterol; HDL-c, high density lipoprotein-cholesterol; ALT, alanine aminotransferase; AST, aspartate aminotransferase. The data were represented as the mean ± S.E.M. *: vs. SF group, *p* < 0.05.

## Data Availability

The datasets generated for this study can be found in the Sequence Read Archive (SRA) database (PRJNA845971).

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
