# Peer review of "Intragastric Safflower Yellow Alleviates HFD Induced Metabolic Dysfunction-Associated Fatty Liver Disease in Mice through Regulating Gut Microbiota and Liver Endoplasmic Reticulum Stress"

_nutrients, 2023, doi:10.3390/nu15132954_

Round 1
Reviewer 1 Report
In this manuscript, Hu and Lyu et al. conducted research to explore the potential protective effects of Safflower Yellow (SY) in both in vitro and in vivo models of fatty liver diseases. The experimental design of the animal model is reasonable, encompassing multiple parameters such as metabolic phenotype, liver phenotype, and metagenomics. However, the use of HepG2 cells, a hepatocellular carcinoma cell line, as a model for studying fatty liver disease could be considered a minor limitation. The authors propose that the protective effect observed may be attributed to the potential regulation of gut microbiota and liver endoplasmic reticulum (ER) stress, both of which are relevant mechanisms in fatty liver disease. Overall, the manuscript is well-written, with minimal errors.
The following are some detailed suggestions that can improve the manuscript.
1. It is unnecessary to reproduce Figure 1F-I in Figure 1J.
2. In Figure 2, authors showed that there is no difference in RER between HFD vs SY-HFD but increase in energy expenditure. Are there any changes in thermogenesis in addition to changes in exercise?
3. Figure 3C, the AST of mice is higher than normal in all groups. Is there any explanation?
4. Figure 3D, label for group in H&E is missing.
5. Are there any differences in liver inflammation levels?
6. In all qPCR results, the gene symbols for mice should be capitalized, but not in all uppercase. For instance, the gene symbol for fatty acid synthase in mice should be "Fasn", whereas for humans, it should be "FASN".
7. In Figure 3G, the authors assessed the expression of hormone-sensitive lipase (gene symbol should be "Lipe") and adipose triglyceride lipase (gene symbol should be "Pnpla2"). However, both genes are primarily expressed in adipocytes to regulate lipolysis and show minimal expression in hepatocytes. Therefore, discussing the expression of these genes in the liver may be irrelevant.
8. The authors investigated ER stress by analyzing the expression of downstream targets of the unfolded protein response (UPR). In addition to the current measurements, it would be valuable for the authors to assess the expression of GRP78 (gene symbol is "Hspa5"), which indicates activation of the ATF6 pathway. Furthermore, the authors could consider observing alternative splicing of Xbp1, which indicates activation of the IRE1a pathway. These additional measurements would provide further insights into the activation of different branches of the UPR.
9. For Figure 4, the label is too small. Particularly the legend for barplot is impossible to see. I suggest the authors change the layout.
10. In Figure 4, the authors indicated an increase in genus of Akkermansia. Did you find any changes in Akkermansia muciniphila which has been shown to be beneficial for obesity patients?
11. Figure 5F, there is no comparison that reached p<0.01, suggest removing it from the legend.
12. Figure 5G, the meaning of the color is not described.
Author Response
We would like to thank you for your careful reading, helpful comments, and constructive suggestions, which has significantly improved the presentation of our manuscript. We have carefully considered all comments from the reviewers and revised our manuscript accordingly. Beside your detailed suggestions, we also supplemented one graph abstract in p.1 line 15 and expressed our thanks to corresponding drawing platform in p.16 line 540-541.
1. It is unnecessary to reproduce Figure 1F-I in Figure 1J.
Response 1: Thank you so much for good suggestions. As what you mentioned, Figure 1J was to reproduce Figure 1F-I. However, the objective of these two parts was different. Figure 1F-I was to present the improvement of glucose metabolism after treating DIO mice by intragastric SY, while Figure 1J aimed to describe the different effects between OGTT and IPGTT in SF, HDF and HDF-SY groups. The AUC of OGTT in the SF significanlty decreased when compared with that of IPGTT, suggesting the glucose disposal in the OGTT in SF group was faster than in the IPGTT as we all known. However, the reduction phenomenon disappeared in HFD group, and the reduction phenomenon was reappeared in DIO mice with SY treatment, indicating that SY treatment may have a significant role in gastrointestinal tract. Therefore, we are more inclined to keep Figure 1J. If you still think that it is unnecessary, we would like to delete it in the next revision.
2. In Figure 2, authors showed that there is no difference in RER between HFD vs SY-HFD but increase in energy expenditure. Are there any changes in thermogenesis in addition to changes in exercise?
Response 2: Considering the Reviewer’s suggestion, we have tested the expression of thermogenesis-related genes Ucp1 and Ppargc1a in subcutaneous adipose tissue (sAT) and brown adipose tissue (BAT). According to our results, no significant increase was observed after SY intervention in sAT, while the expression of Ucp1 and Ppargc1a increased to 1.7-fold and 1.9-fold of DIO control mice respectively in BAT. Corresponding results were supplemented in P.8 line 276-280, Fig.2 (P.8 line 281) and Fig.2 legend (P.9 line 285-288) with highlight.
3. Figure 3C, the AST of mice is higher than normal in all groups. Is there any explanation?
Response 3: We gratefully appreciate for your comment. The normal range of ALT and AST in human is 0-40 U/L. However, it’s different in mice. Our previous researches have reported that the serum levels of ALT in normal wide mice was similar to that in human, while the serum levels of AST in mice with normal diet could reach to about 110 U/L (PMID: 35565866) or 120 U/L (PMID: 36943416). The similar results have also been reported in the researches of Boehm et al. and Widyawaruyanti et al.,which showing the normal range of AST in mice is 100-200 U/L (Reference: DOI: 10.1515/BC.2007.061, DOI: 10.1155/2020/4678634). Men et al. (PMID: 36066106) also reported that the serum levels of AST in wild mice were about 150 U/L.
4. Figure 3D, label for group in H&E is missing.
Response 4: We are very sorry for our negligence of the label. We have supplemented labels in Figure 3D in P.9 line 307. Thank you so much for your careful check.
5. Are there any differences in liver inflammation levels?
Response 5: We gratefully appreciate for your valuable suggestion. We have done RT-qPCR about inflammation-related genes including Tnf and Il1b in liver as well as HepG2 cells and the results were supplemented in P.9 line 301-306 and line P.13 389-392 with highlight. Modification was made in Fig.3 (P.9 line 307) and Fig.6 (P.13 line 395). For liver tissue, intragastric SY significantly reduced the expression of inflammation-related gene Tnf and Il1b by 60.5% and 47.5% respectively. As for HepG2 cells, there was statistically great reduction after 10-100 mg/L SY intervention in TNF. The expression of IL1b showed a decreased trend Although the difference was not statistically significant.
6. In all qPCR results, the gene symbols for mice should be capitalized, but not in all uppercase. For instance, the gene symbol for fatty acid synthase in mice should be "Fasn", whereas for humans, it should be "FASN".
Response 6: We are very sorry for our incorrect writing. According to your comments, we’ve checked our manuscript and modified responding gene symbol in different pages and lines such as P.5 line 204-214, P.8 line 276-280, P.9 line 285-288, P.9 line 301-306, P.10 line 312-317, P.13 line 383-394, P.13 line 398-399, P.14 line 420-427 and P.14 line 438 with highlight.
7. In Figure 3G, the authors assessed the expression of hormone-sensitive lipase (gene symbol should be "Lipe") and adipose triglyceride lipase (gene symbol should be "Pnpla2"). However, both genes are primarily expressed in adipocytes to regulate lipolysis and show minimal expression in hepatocytes. Therefore, discussing the expression of these genes in the liver may be irrelevant.
Response 7: It is really true as Reviewer suggested that Lipe and Pnpla2 show minimal expression in hepatocytes. Therefore, we deleted results associated with these gene expression in live tissue as well as HepG2 cells. And we supplemented the expression of genes related to inflammation such as Tnf and Il1b in P.9 line 301-306 and P.13 line 389-392 with highlight. Modification was made in Fig.3 (P.9 line 307) and Fig.6 (P.13 line 395). For liver tissue, intragastric SY significantly reduced the expression of inflammation-related gene Tnf and Il1b by 60.5% and 47.5% respectively. As for HepG2 cells, there was statistically great reduction after 10-100 mg/L SY intervention in TNF. The expression of IL1b showed a decreased trend Although the difference was not statistically significant.
8. The authors investigated ER stress by analyzing the expression of downstream targets of the unfolded protein response (UPR). In addition to the current measurements, it would be valuable for the authors to assess the expression of GRP78 (gene symbol is "Hspa5"), which indicates activation of the ATF6 pathway. Furthermore, the authors could consider observing alternative splicing of Xbp1, which indicates activation of the IRE1a pathway. These additional measurements would provide further insights into the activation of different branches of the UPR.
Response 8:We gratefully thanks for the precious time the reviewer spent making constructive remarks. Based on expert’s advice, we made RT-qPCR of Hspa5 and Xbp1s which was the potent transcription factor translated from the spliced Xbp1 mRNA. Hspa5 could indicate activation of the ATF6 pathway and Xbp1s is the potent transcription factor translated from the spliced Xbp1 mRNA which could indicate activation of the IRE1a pathway. According to our results, the expression of Hspa5 and Xbp1s decreased significantly after SY intervention (p<0.05). The results were supplemented in P.9 line 303-304 and P.14 line 423-424 with highlight.
9. For Figure 4, the label is too small. Particularly the legend for barplot is impossible to see. I suggest the authors change the layout.
Response 9: Thanks for pointing out this problem. We have changed the legend size for barplot to make the figure clearer to see. The new figure has been shown in P.11 line 359 with highlight.
10. In Figure 4, the authors indicated an increase in genus of Akkermansia. Did you find any changes in Akkermansia muciniphila which has been shown to be beneficial for obesity patients?
Response 10: Thank you for your rigorous consideration. Depommier C et al. have reported supplementation with Akkermansia muciniphila in overweight/obese patients could improve their metabolic status (PMIC: 31263284). Therefore,we have discussed the beneficial effects of Akkermansia at the genus levels in P.14-15 line 447-462. The reference was supplemented in P.18 line 640-641 with highlight.
11. Figure 5F, there is no comparison that reached p<0.01, suggest removing it from the legend.
Response 11: Thanks for your advice. The legend “p<0.01” has been deleted and new figure has been shown in P.12 line 365.
12. Figure 5G, the meaning of the color is not described.
Response 12: We feel sorry for the inconvenience brought to the reviewer. We have attached description “(G) Heatmap of the correlation of metabolic phenotype and differential gut microbiota (n=5 in each group).” of Figure 5G in figure legend in P.12 line 370-372 with highlight.
Reviewer 2 Report
Attach file

Attach file
Author Response
We would like to thank you for your careful reading, helpful comments, and constructive suggestions, which has significantly improved the presentation of our manuscript. We have carefully considered all comments from the reviewers and revised our manuscript accordingly. Beside your detailed suggestions, we also supplemented one graph abstract in p.1 line 15 and expressed our thanks to corresponding drawing platform in p.16 line 540-541.
Point 1: p.1, line 19 (abstract): …. treated with 125mg/kg/d SY for 12 weeks …. Change by …. treated with 125 mg/kg/d SY for 12 weeks ….
Response 1: Thanks for your suggestion. We have changed the description as you suggested with highlight in p.1, line 20 (abstract).
Point 2: p.2, line 46: Carthamus tinctorius L. is a traditional Chinese medicine …. Change by …. Carthamus tinctorius L. is a traditional Chinese medicine ….
Response 2: Thank you for raising this issue. The corresponding modification has been made in p.2, line 47 with highlight.
Round 2
Reviewer 1 Report
Most of the questions were properly addressed. Congretulations.